# New Developments in the Ultrasonography Diagnosis of Gallbladder Diseases

Lara Mencarini [1,2], Amanda Vestito [1], Rocco Maurizio Zagari [1,3] and Marco Montagnani [1,2,*]

1 Department of Medical and Surgical Sciences, University of Bologna, Via Massarenti 9, 40138 Bologna, Italy; lara.mencarini@studio.unibo.it (L.M.); amanda.vestito@aosp.bo.it (A.V.); roccomaurizio.zagari@unibo.it (R.M.Z.)
2 Gastroenterology Unit, IRCCS Azienda Ospedaliero—Universitaria di Bologna, 40138 Bologna, Italy
3 Esophagus and Stomach Organic Diseases Unit, IRCCS Azienda Ospedaliero—Universitaria di Bologna, 40138 Bologna, Italy
* Correspondence: marco.montagnani@unibo.it

**Abstract:** Gallbladder diseases are very common, and their diagnosis is based on clinical–laboratory evaluation and imaging techniques. Considering the different imaging diagnostic tools, ultrasound (US) has the advantage of high accuracy combined with easy availability. Therefore, when a gallbladder disease is suspected, US can readily assist the clinician in the medical office or the emergency department. The high performance of US in the diagnosis of gallbladder diseases is mainly related to its anatomic location. The most frequent gallbladder pathological condition is gallstones disease, easily diagnosed via US examination. Acute cholecystitis (AC), a possible complication of gallstone disease, can be readily recognized due to its specific sonographic features. Additionally, a number of benign, borderline or malignant gallbladder lesions may be detected via US evaluation. The combined use of standard B-mode US and additional sonographic techniques, such as contrast-enhanced ultrasonography (CEUS), may provide a more detailed study of gallbladder lesions. Multiparametric US (combination of multiple sonographic tools) can improve the diagnostic yield during gallbladder examination.

**Keywords:** adenomyomatosis; gallbladder cancer; gallbladder perforation; gallbladder polyps; multiparametric US; MVFI; POCUS; porcelain gallbladder; veno-occlusive disease; xanthogranulomatous cholecystitis

## 1. Introduction

The gallbladder is an easily accessible organ at US examination due to its anatomical location in the upper right quadrant of the abdomen. For this reason, the gallbladder can be thoroughly evaluated via conventional B-mode US, as well as via color Doppler (CD) and CEUS.

When compared to alternative imaging techniques, such as computed tomography (CT) and magnetic resonance imaging (MRI), US has the advantage of a lower cost, easy accessibility and high diagnostic performance in evaluating the gallbladder. Furthermore, US avoids radiation exposure, and this is even more important in pediatric populations and pregnant women.

Gallbladder US can be performed directly at the patient's bed-side, in the emergency department, in hospital wards or at the doctor's office. For this reason, US is time-saving, well tolerated and extremely convenient for the patient and the clinician. Therefore, in the case of suspected biliary disease, US is considered the first-choice imaging technique.

Even more advanced and detailed diagnostic imaging can be obtained via CEUS or by using new sonographic techniques, combined with conventional B-mode US.

Point-of-care ultrasound (POCUS) is an important application of US that allows ready imaging support during the clinical evaluation. Furthermore, US can be easily repeated in the case of gallbladder diseases that require monitoring over time.

## 2. Cholelithiasis

Gallstones are a common disease, and US is the method of choice for diagnosis, with an accuracy as high as 95% [1,2].

The US typical appearance of gallstones is an echogenic focus in the gallbladder lumen that casts a posterior acoustic shadow and changes position according to the variation in patient decubitus (Figure 1) [3]. Stones smaller than 2 or 3 mm may be difficult to visualize, especially if isolated. Gallstones typically produce complete shadowing without reverberation because most of the ultrasound is absorbed by the stone. Occasionally, reverberation artifacts may be seen posterior to calcified stones if they contain gas within fissures [1].

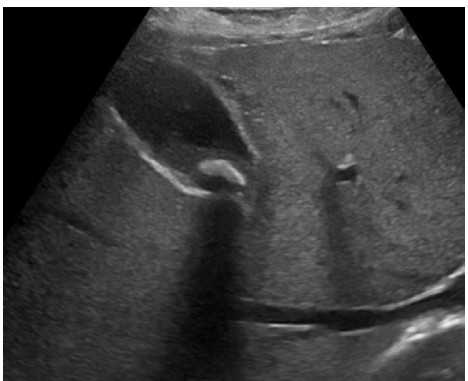

**Figure 1.** Gallbladder lithiasis and sludge. A gallstone can be seen in the bottom of the gallbladder lumen, with the typical sonographic appearance of an echogenic focus casting an acoustic shadow. Biliary sludge (echoic non-shadowing material with indefinite morphology) surrounds the gallstone, forming a horizontal fluid–fluid level.

When the lumen of the gallbladder is completely filled with stones, the usual finding is a highly reflective gallbladder fossa, known as the "wall-echo-shadow" (WES) triad or "double-arc-shadow sign" [4]. The WES complex consists of two parallel arcuate hyper-echoic lines separated by a thin hypoechoic space and distal acoustic shadowing. The most external hyperechoic arc represents the anterior side of the gallbladder wall. The hypoechoic space in between is a small film of bile separating the gallbladder wall from the stones. Alternatively, the hypoechoic layer could represent a portion of the gallbladder wall. Finally, the deeper hyperechoic line represents the gallstones with an acoustic shadow that masks the rest of intraluminal stones and the posterior gallbladder wall [5].

On CD, the highly reflective surface of the stones, particularly in case of cholesterol stones, can produce the typical "twinkling artifact", characterized by a mosaic of colored pixels posterior to the stone [6].

Finally, US is very useful in the setting of biliary colic. In particular, POCUS is increasingly employed in the emergency room for the evaluation of patients with upper right quadrant pain [7], mainly to confirm the suspicion of the presence of gallstones or a different gallbladder disease.

## 3. Gallbladder Sludge

Gallbladder sludge is made of mucins, glycoproteins, calcium and pigments. This environment facilitates cholesterol crystallization and calcium bilirubinate precipitation, predisposing patients to gallstones development [8].

Typically, biliary sludge is a slight sonographic finding, which appears as non-shadowing echoes with an indefinite shape that tends to layer in the most declivous portion of the

gallbladder [1]. Biliary sludge forms a horizontal level that moves slowly, according to changes in a patient's decubitus (Figure 1) [1,2].

Occasionally, aggregated sludge may appear as a static, variably echogenic and intra-luminal mass, without acoustic shadows or internal vascular signals, in close proximity to the gallbladder wall ("sludge ball") or as a polypoid mass ("tumefactive sludge") [2,3]. A gallbladder completely filled with sludge may be isoechoic with the adjacent liver, leading to the so called "hepatization of the gallbladder" [3] In this setting, it can be difficult to distinguish biliary sludge from polypoid lesions, if not from gallbladder carcinoma (GBC) [9,10].

CD examination can be useful in differentiating biliary sludge from a solid mass. In particular, the presence of CD signals can be considered a reliable indicator of a malignant lesion, while the lack of signal does not exclude malignancy [11].

In the case of tumefactive sludge, some authors recommend repeating the US examination after a short interval of time (from 1 day to 2 weeks), especially in patients on prolonged fasting, after resuming normal habits [9].

CEUS can greatly improve the diagnostic confidence in the differential diagnosis between sludge and mass-forming lesions, especially when CD signals are not detectable. Indeed, sludge typically does not show any kind of enhancement at any contrast phase due to the absence of vascularization, with an accuracy of 100% [12–18].

The pseudo-enhancement of gallbladder sludge, caused by an artifact due to the nonlinear propagation of US through microbubbles, has been reported in a single case report [19].

According to some authors, new techniques for the detection of small vessel flow (MVFI, Microvascular Flow Imaging, and SMI, Superb Microvascular Imaging) can help to differentiate tumefactive sludge from the solid lesions of the gallbladder [20].

### 4. Gallbladder Hydrops

Gallbladder hydrops, sometimes denoted as mucocele, is generally defined as a distended gallbladder filled with mucoid fluid due to the prolonged impaction of a stone in the cystic duct or the gallbladder neck. This condition is associated with the interruption of gallbladder filling and the reabsorption of the endoluminal bile [21]. In turn, this can evolve into acute cholecystitis due to bacterial growth [22]. Additional causes of gallbladder hydrops are obstructing polyps and tumors, congenital strictures, ascariasis infestation or external compression by enlarged lymph nodes [23]. A specific form of gallbladder hydrops is observed in children during the acute phase of Kawasaki disease [23].

US shows a distended gallbladder with intraluminal clear fluid and possibly gallstones or biliary sludge in the gallbladder neck or the cystic duct [21]. A distended gallbladder is defined as an axial diameter > 4–5 cm [21,23]. Some authors also take into account a longitudinal diameter > 10 cm [22].

### 5. Acute Cholecystitis

In most patients, the acute inflammation of the gallbladder wall is caused by gallstones, impacted in the cystic duct or in the gallbladder neck (calculous cholecystitis).

Recently, the Tokyo guidelines proposed specific criteria for the diagnosis of AC, providing a pivotal role for imaging studies [24]. In particular, US is recommended as the imaging examination for the diagnosis of AC due to its high sensitivity and specificity (88% and 80%, respectively) [24,25].

In patients with suspected AC, POCUS can be readily performed at admission into the Emergency Department. In particular, the presence of specific sonographic findings (gallstones, sludge, gallbladder wall thickening, pericholecystic fluid), especially if combined, has a good sensitivity and a high specificity for the diagnosis of AC [7,26].

Besides the presence of gallstones or biliary sludge, the sonographic criteria for AC include gallbladder wall thickening (>3 mm, in fasting condition) with a layered appearance, gallbladder enlargement (longitudinal diameter > 8 cm, axial diameter > 4 cm),

pericholecystic fluid and a positive sonographic Murphy sign [2]. The sonographic Murphy sign (tenderness elicited by the compression of the transducer over the gallbladder) is considered more accurate than a conventional physical examination. In particular, the presence of cholelithiasis combined with a positive sonographic Murphy sign seems to be the most specific diagnostic finding in AC [1,27,28]. Of note, it may not always be possible to evaluate for the sonographic Murphy sign (e.g., in unresponsive patients or after the administration of pain medication) [1].

An isolated sonographic finding is not per se sufficient, while, in the proper clinical setting, concurrent multiple US features are highly accurate for the diagnosis of AC [3].

Sometimes, the adjacent liver parenchyma may show findings suggestive of diffuse edema, such as a hypoechoic aspect, possibly with prominent echogenic portal triads, known as "starry-sky" appearance (Figure 2) [27,29].

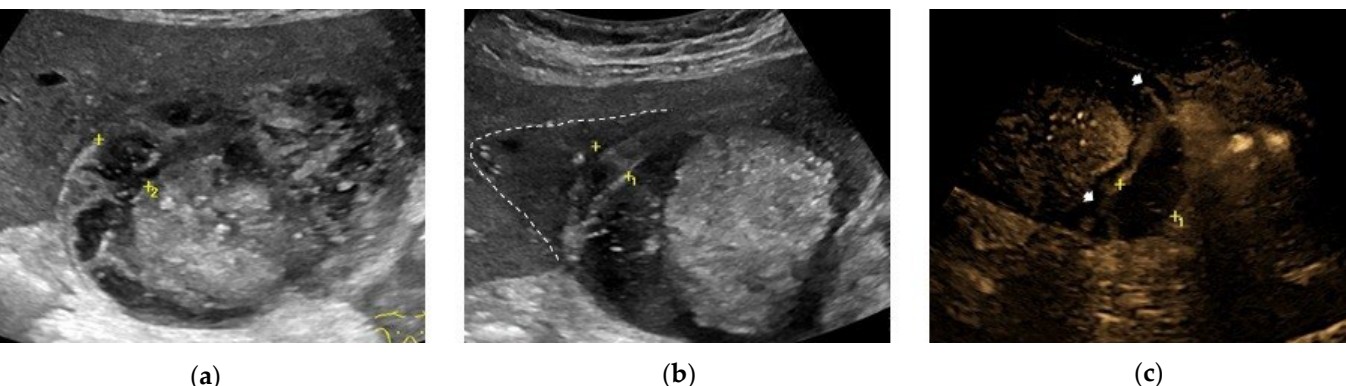

(**a**)          (**b**)          (**c**)

**Figure 2.** Complicated acute cholecystitis, leading to gangrenous cholecystitis (GC), with irregular gallbladder wall thickening with multiple striations and alternating hypo/hyperechoic bands, (**a**,**b**) and coexisting empyema (abundant dishomogeneous echogenic material occupying most of the gallbladder lumen, without posterior acoustic shadows) (**a**,**b**). Slightly hypoechoic pericholecystic liver parenchyma (delimited by the dotted line) corresponds to concurrent hepatic edema (**b**). MVFI shows perfusion defects (arrows) of the inner layer with an irregular and thickened gallbladder wall, without the vascular signal in the underlying middle portion, due to edema and tissue necrosis (**c**).

Hyperemia of the gallbladder wall can be detected via CD evaluation in AC [10,24]. MVFI has also been proposed to improve the detection of parietal blood flow [20,30].

In course of AC, CEUS shows fast, homogeneous and intense arterial uptake of the thickened gallbladder wall compared to the liver parenchyma, [12,14], with a slight hypo-enhancement in the late phase [18,31]. An indirect sign of AC is the hyper-enhancement of the adjacent liver tissue (pericholecystic hepatitis) with respect to the rest of the liver parenchyma [29,32,33]. For the same reason, the shear wave elastography (SWE) and shear wave dispersion slope (SWD) of the pericholecystic liver tissue may be increased in a course of AC [34].

## 6. Acute Acalculous Cholecystitis

Acute acalculous cholecystitis (AAC) is an acute inflammatory disease of the gallbladder in the absence of intraluminal gallstones. This clinical entity accounts for 5–10% of AC cases, and it is associated with high morbidity and mortality [35].

Usually, it is found in special clinical settings, e.g., total parenteral nutrition, bone marrow transplant, severe trauma, burns, critical illness, cardiac surgery with cardiopulmonary bypass, immunodeficiency, immunosuppression, diabetes mellitus, systemic vasculitis and COVID-19 [36]. In these clinical contexts, gallbladder stasis and ischemia can occur, with the subsequent development of AAC. Sometimes, AAC is due to a primary infection, especially by opportunistic pathogens in a course of AIDS (e.g., *Cryptosporidium*, Cytomegalovirus or Microsporidia) [37]. The obstruction of the cystic duct by biliary cancer, extrinsic inflammation, lymphadenopathy or metastasis can also lead to AAC [25,31]. No-

tably, AAC represents the most frequent form of acute cholecystitis in children. It usually develops in the setting of infectious or parasitic diseases (in particular Epstein–Barr virus and hepatitis A infection), systemic vasculitis (e.g., Kawasaki disease and polyarteritis nodosa) or congenital malformations of the gallbladder and biliary tract [36].

Sonographic features suggestive of AAC are nearly the same for AC, except for the absence of gallstones in the first condition [29,31]. In AAC, the sensitivity and specificity of US reach 92% and 100%, respectively [35,38,39]. In cases of high clinical suspicion but unspecific US picture, a serial sonographic examination can be easily performed to monitor for the development of AAC [40,41].

## 7. Acute Complicated Cholecystitis

The nosological pictures of acute complicated cholecystitis are typically represented by gangrenous cholecystitis (GC), gallbladder perforation, pericholecystic abscess, emphysematous cholecystitis (EC), gallbladder empyema (GE) and Mirizzi syndrome. These complications often coexist in the clinical practice and can be recognized at the US examination.

### 7.1. Gangrenous Cholecystitis

GC is defined as a form of AC with ischemia and secondary necrosis of the gallbladder wall. GC is the most common complication of AC, with a prevalence up to 20%, particularly in patients with risk factors, such as older age, diabetes mellitus and cardiovascular disease [42]. The early recognition of GC is important because it is associated with increased morbidity and mortality [43].

The most common sonographic finding in GC is an irregular gallbladder wall thickening, [44,45] characterized by multiple striations with alternating hypoechoic or hyperechoic bands (Figure 2) [25]. This sonographic pattern is due to the presence of intramural hemorrhage or micro-abscesses [3,39]. Intraluminal membranes, formed by strands of fibrinous exudate and desquamated or "denuded" mucosa, are considered a more specific finding in GC, although they are less common (Figure 3) [3,39,44,45].

Of note, the sonographic Murphy sign is negative in about two thirds of patients, presumably due to ischemic denervation [27,39,44]. According to some authors, the presence of hyperechoic pericholecystic fat is indicative of flogistic involvement and a specific finding for early gangrenous evolution [46].

Some authors suggest that CD examination can help to diagnose GC, showing a focal decrease in the wall perfusion [47].

At CEUS examination, the hallmark of GC consists of the discontinuous or irregular enhancement of the gallbladder wall due to perfusion defects in the presence of gallbladder wall necrosis [32,48,49]. Similarly, intraluminal membranes do not show a vascular signal [14]. In a comparative study, CEUS had a high sensitivity and specificity in detecting GC, referred to in successive surgical and pathological findings. On the other hand, the same authors reported a few cases with a focal wall defect secondary to perforation in AC without concomitant gangrene [49].

In the presence of edema and necrosis, MVFI can show the alterations to the gallbladder wall already described via CEUS (Figure 2).

### 7.2. Gallbladder Perforation

Gallbladder perforation is caused by transmural necrosis, usually in the setting of acute cholecystitis [29]. In particular, perforation occurs in course of gangrenous cholecystitis or, rarely, in course of acute but not gangrenous cholecystitis [49]. Perforation occurs in about 10% of acute cholecystitis cases and is usually localized in the fundus because of its relatively poor blood supply by the terminal vascular branches of the cystic artery.

According to the Niemeier classification, US can describe the initial rupture of the gallbladder wall into the peritoneal cavity, the subsequent development of a pericholecystic or liver abscess and, finally, a bilio-enteric fistula [50].

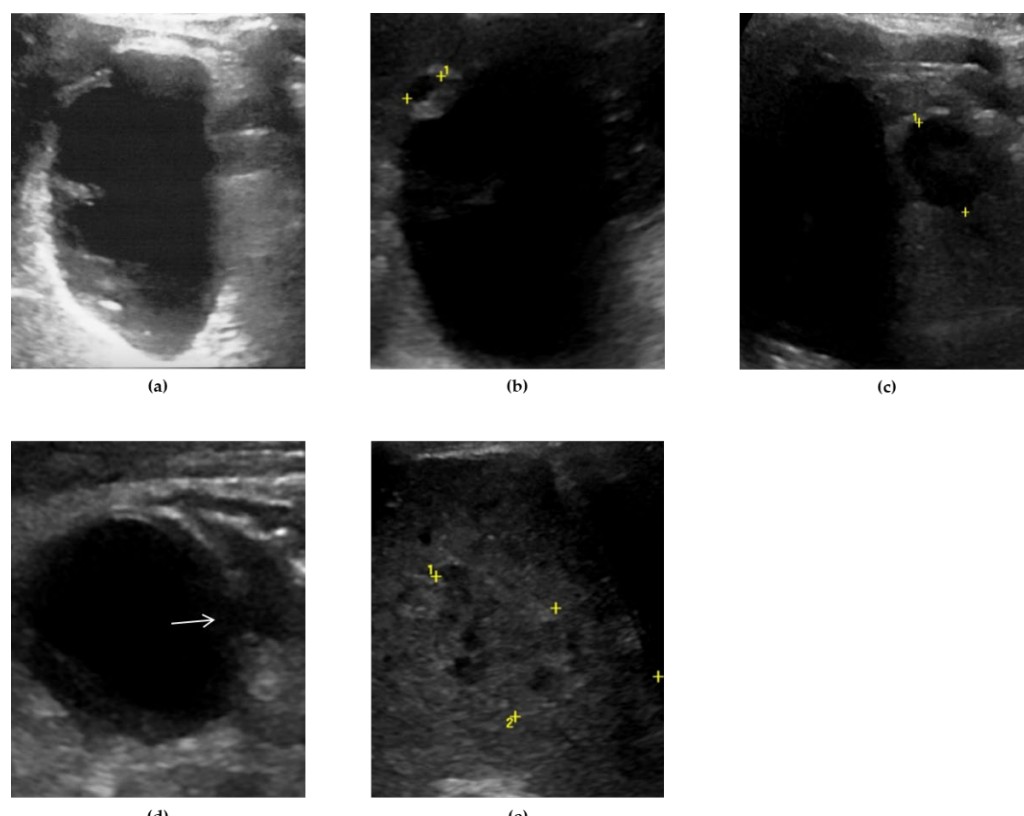

**Figure 3.** Complicated acute cholecystitis, leading to GC and gallbladder perforation. Gallbladder walls are irregularly thickened with an echogenic strand departing from the gallbladder wall and projecting into the lumen (intraluminal membrane: fibrinous exudate and desquamated mucosa). Microlithiasis and biliary sludge can be seen at the bottom of the gallbladder lumen (**a**). At the level of the body, an intramural anechoic collection of the gallbladder wall represents a microabscess (calipers) (**b**). On the left side of the gallbladder, a pericholecystic collection (calipers) (**c**) communicates with the gallbladder lumen through a full thickness parietal defect (hole sign) (arrow) (**d**). Dishomogeneous pericholecystic liver parenchyma (delimited by calipers) corresponds to concurrent phlegmonous reaction/microabscesses of the V hepatic segment (**e**).

At US examination, the hallmark for the diagnosis of gallbladder perforation consists of the "hole sign", present in 45–70% of cases [25] and characterized by the full-thickness defect of the gallbladder wall (Figures 3 and 4) [39,45]. Defects in the gallbladder wall are usually focal and small. However, in case of perforation secondary to infectious necrosis, a large defect may be observed [51]. The full-thickness disruption of the gallbladder wall can be further highlighted via CEUS examination [18,52].

An indirect but specific sign of gallbladder perforation consists of the detection of gallstones outside the gallbladder lumen, typically in the peritoneal cavity [29,39]. Besides the above-mentioned typical features, additional signs may be found. Some authors suggest that in case of a thicker gallbladder, wall perforation is more frequent [25]. Pericholecystic fluid or increased fat echogenicity due to mesenteric reaction may be observed adjacent to the site of perforation [3,29,39].

According to some authors, MVFI may help to detect focal areas of decreased vascular perfusion in the gallbladder wall [20,53].

A careful CD analysis can demonstrate the to-and-fro signal of gallbladder content across the wall defect [54].

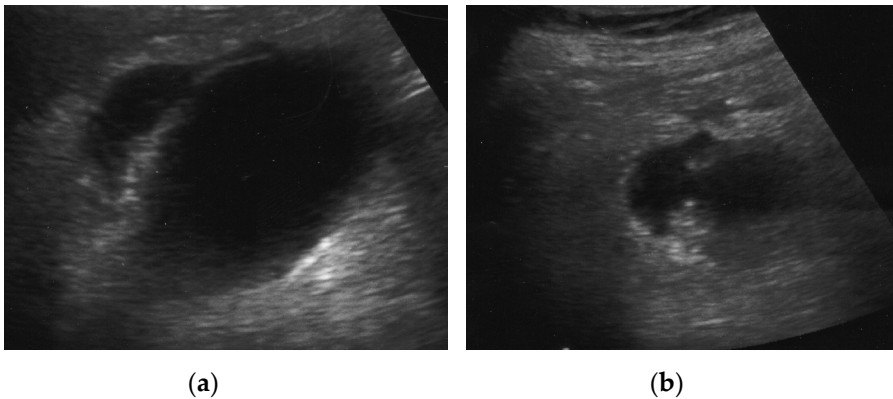

(**a**)                       (**b**)

**Figure 4.** Gallbladder perforation. US B-mode evaluation shows a pericholecystic collection in the setting of acute cholecystitis (**a**). A different scan shows a full thickness defect in the gallbladder wall (hole sign) (**b**).

### 7.3. Pericholecystic Abscess

In case of a subacute process, gallbladder perforation generally results in pericholecystic abscess or, rarely, in a liver abscess (Figure 3) [44].

CEUS is very useful in order to exactly define the presence of pericholecystic collections and abscesses [14,18]. In case of pericholecystic phlegmon CEUS, findings are variable. Usually, the lesions may show hyper-enhancement in the arterial phase, while in a later phase, they may show non-enhancing foci due to the liquefaction process [55]. The typical CEUS appearance of a mature abscess consists of a hyper-enhanced peripheral rim with a central area completely devoid of vascular signal [55]. Sometimes, the abscess displays a honeycomb pattern with multiple septa and areas of nonenhancement [12,52] based on the amount and distribution of the necrotic material [14]. The hyper-enhanced portions correspond to the capsule or the septa provided with vascularized tissue [55]. In some cases, CEUS demonstrates a hypo-enhancement in the peripheral rim or septa of hepatic abscess during the portal venous phase. At the same time, the surrounding liver parenchyma may show diffuse hyper-enhancement and, possibly, a subsequent signal decrease in the late phase [55].

Recently, in cases of in-acute complicated cholecystitis treated via percutaneous cholecystostomy, intracavitary CEUS has been described. The technique refers to the administration of an ultrasound contrast agent inside a physiological or pathological body cavity through a drainage catheter or puncture needle [32]. In case of percutaneous cholecystostomy, an ultrasound contrast agent can be injected directly in the gallbladder lumen in order to verify the correct position of the drainage catheter to detect any possible leak beyond the organ that could indicate a perforation [56]. Similarly, the direct injection of ultrasound contrast agent inside an abscess can be used to confirm the successful placement of the drainage catheter. Moreover, this technique can be useful to highlight the size of the abscess and the presence of septa, communicating compartments inside the abscess or the development of fistulas [57,58]. Intracavitary CEUS can be repeated over time to monitor the evolution of gallbladder wall defect and the pericholecystic abscess in order to improve the therapeutic management.

### 7.4. Emphysematous Cholecystitis

EC is characterized by the presence of gas within the gallbladder wall or lumen in a course of AC in the absence of anomalous communication between the biliary system and the gastrointestinal tract (e.g., previous sphincterotomy or biliary enteric anastomosis) [29,59]. It occurs in approximately 1 to 3% of cases AC [45], particularly in elderly male diabetic patients [29,59]. EC usually results from thrombosis or the occlusion of the cystic artery, leading to ischemic necrosis of the gallbladder wall, and only two thirds of patients have gallstones. In turn, this leads to the proliferation of gas-forming organisms (e.g., *Klebsiella,*

*Clostridium* or *Escherichia coli*) [29,59–61]. Frequently, EC leads to gangrene, perforation and other complications [3,45,60].

Sonographic findings of EC are different according to the amount and localization of gas [45,61]. Gas bubbles can be intraluminal, intramural or in the pericholecystic tissue [45,59]. In case of a small amount of gas, intraluminal bubbles appear as highly reflective punctate echoes, associated with distal dirty shadowing, known as a ring-down or comet-tail artifact [29,59]. Sometimes, US examination displays intraluminal gas bubbles rising from the dependent up to the nondependent portions of the gallbladder cavity ("champagne sign" or "effervescent gallbladder") [62,63]. Of note, free bubble air in EC should be differentiated from the rare condition of gas-containing gallstones [64]. In the case of a large amount of intraluminal bubbles, a wide linear or curvilinear hyperechoic band is shown at the top of the gallbladder lumen. In this case, a typical reverberation pattern known as the powder snow-like posterior echo prevents the visualization of the underlying gallbladder wall [39,45,61]. Notably, a gallbladder filled with gas can be confused with a gas-filled duodenum: US examination with intercostal scans helps to make the correct diagnosis [61]. A gallbladder filled with gas can resemble a porcelain gallbladder or a highly contracted gallbladder filled with gallstones [59,61]. However, in the case of gallstones or a porcelain gallbladder, the echogenic line is usually sharp and always associated with posterior acoustic shadow [61]. In case of doubt, changing the patient's position determines intraluminal gas movement, suggesting the diagnosis of EC. In case of intramural gas, US examination may show multiple areas of high reflectivity with distal reverberations or, alternatively, a single bright ring of hyper-reflective echoes within the thickened gallbladder wall [3,59]. The detection of pericholecystic gas bubbles suggests that EC has led to gallbladder perforation [59].

### 7.5. Gallbladder Empyema

GE, also known as suppurative cholecystitis, occurs when purulent material accumulates within a distended gallbladder in the setting of AC due to persistent obstruction of the cystic duct and stasis of contaminated bile. GE typically occurs in diabetic patients and can determine gallbladder perforation and sepsis [65].

Typically, at US examination, the gallbladder is distended and contains intraluminal echogenic material [66] without posterior acoustic shadows, lying in the dependent portion of the lumen. The echogenic material moves as the patient's position change, usually resembling biliary sludge (Figure 2) [60]. Additional sonographic findings of GE are gallbladder wall thickening and pericholecystic fluid [66,67]. Rarely, the presence of intraluminal air has been reported, suggesting infection by anaerobic pathogens such as *Clostridium* or *Bacteroides* [67].

### 7.6. Mirizzi Syndrome

Mirizzi syndrome is a rare complication of cholelithiasis (it affects up to 5.7% of patients undergoing surgery for cholelithiasis), [68] occurring when an impacted gallstone in the gallbladder neck, infundibulum or cystic duct causes the extrinsic compression of the common hepatic duct [29,39]. A cholecysto-biliary fistula may eventually develop. Based on the presence and extent of cholecysto-biliary fistula, four types of Mirizzi syndrome have been described [29,39,69]

Mirizzi syndrome is generally suspected in a course of abdominal US, which shows the dilatation of the biliary system above the level of the gallbladder neck or cystic duct, in presence of one or more stones. Distally to the stenosis, the common bile duct maintains a normal caliber [29,39,70]. Sonographic findings may include gallbladder hydrops, contracted atrophic gallbladder or, in case of concomitant AC, gallbladder wall thickening. US sensitivity is reported to be up to 50%, and in one study even 77% [70–72]. The presence of malignancy must be ruled out before making a therapeutic decision.

## 8. Chronic Cholecystitis

Chronic cholecystitis is usually associated with gallstones and refers to chronic inflammatory cell infiltration and fibrosis of the gallbladder wall. It is the consequence of a mild, long-standing gallbladder inflammation. Around 5–10% of chronic cholecystitis cases develop in the absence of gallstones, although some authors have reported this condition in up to 25% of cases [2,60]. Most of patients are asymptomatic, although some patients report a history of recurrent acute cholecystitis or biliary colic. Chronic cholecystitis can evolve into AC and GBC [29].

At US examination, the gallbladder is typically contracted, with uniform circumferential wall thickening, characterized by a preserved two-layer structure [2,25]. Rare forms of segmental chronic cholecystitis have been reported [73].

In most cases of chronic cholecystitis, CEUS examination shows hyper-enhancement of the gallbladder wall in the arterial phase, [18] without the typical features of malignant lesions (see the specific section) [17,74]. In case of biliary symptoms, the management of chronic cholecystitis is cholecystectomy, possibly in a symptom-free interval [29,74].

Besides the typical presentation, different subtypes of chronic cholecystitis may be found, namely xanthogranulomatous cholecystitis (XGC), porcelain gallbladder, IgG4-associated cholecystitis (IgG4-CC) and some other rarer forms (hyalinazing cholecystitis, eosinophilic cholecystitis) [25].

### 8.1. Xanthogranulomatous Cholecystitis

XGC is characterized by the accumulation of lipid-filled macrophages and mixed inflammatory cell infiltrates in the gallbladder wall [25,75]. It is a rare entity, with a prevalence ranging from 1 to 2% in cholecystectomized patients in Western countries and up to 9% in India [76]. Xanthogranulomatous reaction is thought to originate from the extravasation of bile into the wall and the Rokitansky–Aschoff sinuses (RAS). Occasionally, XGC has been found in association with gallbladder adenomyomatosis (GA) [25].

At US examination, the presence of well-defined hypoechoic nodules or bands within the thickened gallbladder wall represents the xanthogranulomatous reaction documented in histopathologic analysis, and it is considered highly suggestive of XGC [25,75,77]. However, the hypoechoic nodules can be misdiagnosed with gallbladder intramural abscess [75] or RAS [3]. The gallbladder wall is focally or diffusely thickened with a preserved mucosal layer [25,60]. Intraluminal gallstones are present in about 85% of cases, suggesting a potential role in the pathogenesis of XGC [78]. Additional sonographic findings may include hyperdense intramural nodules, pericholecystic fluid, sludge and the hyperechoic appearance of the adjacent liver parenchyma [60]. In fact, in the most severe cases, xanthogranulomatous inflammation can extend from the gallbladder to the adjacent structures such as liver, bowel and stomach, resulting in adhesion, perforation, abscess and fistula, possibly detectable via US examination [75].

The diagnosis of XGC can be improved by means of high-resolution ultrasound (HRUS), defined as the use of a 7–10 MHz linear probe combined with a 3–5 MHz convex probe. The high-frequency linear transducer allows for the higher imaging resolution of the gallbladder wall, which can result in a clearer and more accurate visualization of XGC features [2,79–81].

When XGC is suspected, an important issue is the differential diagnosis with GBC, both in the pre- and intraoperatory settings, especially in cases of severe proliferative fibrosis of the gallbladder and surrounding tissues [82]. Occasionally, XGC has been found in association with GBC [83]. Therefore, the sonographic suspect of XGC warrants a cholecystectomy [29].

According to some authors, CEUS may help to diagnose XGC by detecting a continuous inner gallbladder wall enhancement in the arterial phase, with late hypo-enhancement. The diffuse thickening of gallbladder wall and hypoechoic nodules are also highlighted via CEUS [84,85].

### 8.2. Porcelain Gallbladder

Porcelain gallbladder is a rare variant of chronic cholecystitis, diagnosed in less than 1% of cholecystectomy specimens, characterized by the calcification of the thickened gallbladder wall, resulting from long-standing inflammation [1,25]. Its name derives from the brittle consistency of the gallbladder. [1] The histological hallmark of porcelain gallbladder consists of mural calcification, ranging from focal plaques within the mucosal layer to a broad and continuous band that includes and replaces the muscular layer. [1] These pathological changes can involve the entire gallbladder or may be confined to part of the organ [1]. Of note, intraluminal gallstones are present in more than 95% of cases of porcelain gallbladder [86]. Previous studies reported a cancer risk up to 60% in porcelain gallbladder, while recent data have found a much lower association (2–3%) [2,25]. Interestingly, the increased risk may be confined to patients with selective mucosal calcification or incomplete mural calcification [87,88].

At US examination, porcelain gallbladder typically appears as a curvilinear or linear hyperechoic structure with a wide posterior acoustic shadow in the gallbladder fossa. This variant corresponds to complete intramural calcification and must be differentiated from a gallbladder completely filled with stones [25]. In cases of selective mucosal calcification, two different sonographic patterns are described: scattered punctuate echoes with acoustic shadow within the gallbladder wall and, alternatively, a biconvex, curvilinear hyperechoic structure with variable acoustic shadowing in the gallbladder fossa [1]. The US appearance of the porcelain gallbladder may be similar to EC, but the clinical setting is considerably different [2].

In the setting of porcelain gallbladder, CEUS can characterize a coexisting mass with signs of malignancy (see the specific section).

### 8.3. IgG4-Related Cholecystitis

IgG4-CC is an emergent organ manifestation of IgG4-related disease, and only a few cases are reported in the literature [89].

At US examination, the main finding of IgG4-CC is the thickening of the gallbladder wall, which can be diffuse or localized [25]. Recently, a further classification of IgG4-CC wall thickening has been proposed in Japan due to the wider diffusion of IgG4-related diseases [89].

Sometimes, it can be challenging to differentiate IgG4-CC from GBC [89,90]. For this reason, the important role of CEUS in IgG4-CC is to exclude malignancy (see the specific section) [89].

## 9. Hyperplastic Cholecystoses

Hyperplastic cholecystoses refer to different conditions characterized by lipid accumulation (cholesterol and triglycerides) in the gallbladder wall. The two main variants of gallbladder cholecystoses are GA and gallbladder cholesterolosis.

### 9.1. Gallbladder Adenomyomatosis

GA is relatively common, observed in 2% to 9% of cholecystectomy specimens [91–93]. At the histological examination, GA is characterized by gallbladder wall thickening with epithelial proliferation, muscular hypertrophy and RAS (mucosal invaginations into the thickened muscularis propria) [31]. According to its distribution, GA can be classified into focal, segmental (or annular) and diffuse types [2]. Sometimes, these features may coexist (Figure 5). The focal variant, also called adenomyoma, is the most frequent, and the gallbladder fundus is most commonly involved. Occasionally, a focal adenomyoma may appear as an intraluminal polypoid projection. The segmental type affects a ring of the gallbladder body with luminal narrowing, resembling an "hourglass" or a "waist-like" appearance [94]. Some authors consider segmental GA to be a precancerous condition [95,96]. The diffuse type is the less common form and involves the entire organ.

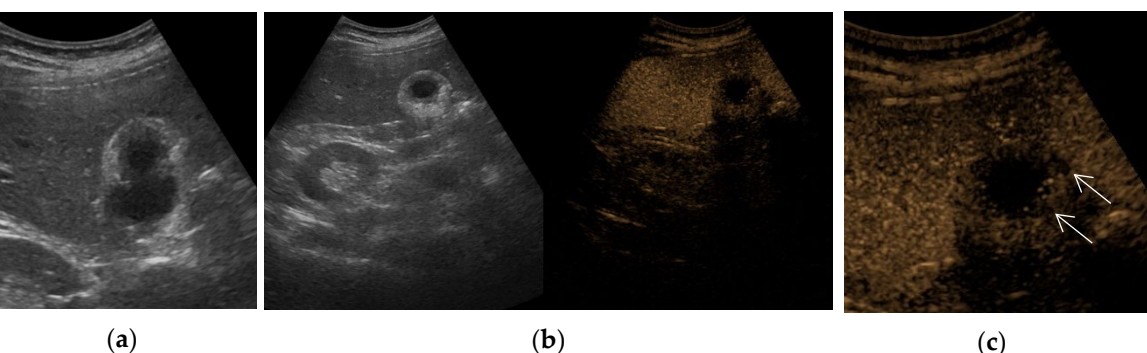

|  |  |  |
|:---:|:---:|:---:|
| (**a**) | (**b**) | (**c**) |

**Figure 5.** Mixed type gallbladder adenomyomatosis showing body and fundic wall thickening coexisting with "hourglass" or "waist-like" appearances (**a**). CEUS examination shows a "mouth-eaten" enhancement in the arterial phase without wash-out in the late phase, where RAS appears in the form of avascular spaces (arrows) (**b**,**c**).

At US examination, the presence of RAS in a thickened gallbladder wall strongly suggests GA. However, RAS are not always detected, given their small size, ranging from 1 to 10 mm. Notably, the use of HRUS can highlight the presence of RAS, which appear as small anechoic spaces within the gallbladder wall [97]. When RAS are filled with sludge, small stones or cholesterol crystals, their internal structure become echoic. In particular, cholesterol crystals inside RAS appear as echogenic mural spots and act as highly reflective surfaces that generate posterior reverberation artifacts (vertical "V-shaped" echoes, known as "ring down" or "comet tail" artifact). Similarly, CD examination can show a persistent signal composed of a rapidly changing mixture of red and blue color (twinkling artifact) [98,99].

CEUS can be very useful in the diagnosis of GA by increasing the detection of RAS, which do not show vascular enhancement. Typically, in the arterial phase the thickened gallbladder wall shows a "moth-eaten" pattern (Figure 5) [14]. In the venous phase, the thickened wall can display a slight hypo-enhancement [12]. Importantly, CEUS can help exclude malignant lesions (see the specific section) [13,100–102].

Cholecystectomy can be considered in cases of biliary symptoms or when potential malignant evolution is suspected (e.g., in case of segmental adenomyomatosis) [103].

### 9.2. Gallbladder Cholesterolosis

Gallbladder cholesterolosis is a form of hyperplastic cholecystosis characterized by lipid accumulation (cholesterol, cholesterol esters and triglycerides) in the macrophages of the lamina propria. It may involve the gallbladder in a focal or diffuse form. The prevalence ranges from 9% to 26% of cholecystectomy specimens [104].

At US examination, lipid accumulation appears as bright hyperechoic foci within the gallbladder wall, which may show a comet tail artifact. With time, they can be covered by the normal gallbladder epithelium, with the development of cholesterol polyps. The diffuse form of gallbladder cholesterolosis corresponds to the so-called "strawberry gallbladder", a pathologic finding characterized by a bright red mucosa with slightly raised interposed areas of yellow lipid aggregates at gross examination [91,105].

## 10. Gallbladder Polyps

Gallbladder polyps are mucosal projections with varying shapes and sizes, rising from the gallbladder wall and protruding into the lumen [106,107]. Gallbladder polyps are a common finding at US examination (from 1.5% to 4.5%), with an even higher prevalence at histological analysis (up to 13.8%) [106]. They are classified into benign (neoplastic or non-neoplastic) and malignant lesions [2]. Cholesterol polyps are the most common type, followed by adenomyomas, inflammatory polyps and adenomas. Leiomyomas, fibromas, lipomas and heterotopic mucosa have also been described [108]. Adenocarcinoma is the

most common type of malignant polyp. Less common forms are lymphoma, sarcoma, mucinous cystoadenoma, squamous cell carcinoma, adenoacanthoma and metastases from different malignant lesions [106,107].

### 10.1. Cholesterol Polyps

Cholesterol polyps are the most common form of benign gallbladder polyp and represent a morphologic variant of gallbladder cholesterolosis [3,106]. At US examination, they typically appear as multiple, small (usually 1–2 mm, rarely up to 10 mm), round-shaped and intraluminal hyperechoic masses with smooth contours, fixed to the gallbladder wall regardless of positional change, without any acoustic shadow. Their stalks are rarely visible, giving a typical appearance known as the "ball on the wall" sign [91].

### 10.2. Inflammatory Polyps

Inflammatory polyps result from granulation and fibrous tissue secondary to chronic inflammation. They are often associated with chronic cholecystitis and gallstones. At US examination, they typically appear as small (5–10 mm), sessile or peduncolated polyps, without comet tail artifacts, acoustic shadows or any other specific sonographic features [108,109].

### 10.3. Adenomas

Adenomas are the most common benign neoplastic polypoid lesions of the gallbladder, accounting for a relatively low overall prevalence (4% of benign gallbladder polyps). US examination typically shows a single, sessile or pedunculated mass, variable in size (from 5 to 20 mm), with detectable vascular flow at CD and CEUS examination. In almost 50% of cases, gallbladder adenomas are associated with intraluminal gallstones [106,108].

Biliary papillomatosis is a rare condition characterized by the presence of multiple adenomatous foci, diffuse or confined to a specific segment of the biliary tract. When the gallbladder is involved, its internal surface is interested by multiple small polyps (multicentric papillomatosis) [110].

Although the progression from adenoma to carcinoma is not definitively proved, the polyp size is considered to be directly related to the risk of gallbladder malignancy [108]. Some authors describe a cancerization risk of 128.2 per 100,000 person-years in the case of an adenomatous polyp more than 10 mm in size [111].

### 10.4. Sonographic Differentiation and Characterization of Gallbladder Polyps

US has the highest sensitivity (84%) and specificity (96%) in the detection of gallbladder polyps, and it is the most used imaging technique in the diagnosis, characterization and follow-up of these lesions (Table 1) [31,112].

**Table 1.** Differential diagnosis between cholesterol and adenomatous gallbladder polyps based on sonographic features.

| Differential Diagnosis | Cholesterol Polyps | Adenoma |
|---|---|---|
| Number | Multiple | Single |
| Size | <10 mm (usually 2–4 mm) | Usually >10 mm (range 5–20 mm) |
| Echogenicity | Iso-/hyperechoic, with internal hyperechoic foci | Hypoechoic, sometimes with internal hypoechoic foci |
| Surface | Smooth | Sessile or lobulated contours |
| Stalk | Thin, usually absent | Wide |
| Vascular flow at CD evaluation | Absent | Present (not always detectable) |
| Arterial flow (CEUS) | Isoenhancement | Hyper-enhancement |

HRUS proved to be more accurate for the diagnosis and characterization of gallbladder polyps than conventional US [81].

CEUS can highlight some features of gallbladder polyps (e.g., size, morphology, and stalk). Cholesterol polyps typically show isoenhancement in the arterial phase, while most adenomatous polyps show homogeneous hyper-enhancement. Considering the microvascular pattern, most authors do not find substantial differences between cholesterol and adenomatous polyps. A single study described a different behavior in the arterial phase for cholesterol polyps (dotted pattern) and adenomatous polyps (linear pattern) [113]. In the late phase, both cholesterol and adenomatous polyps tend to be isoechoic to the surrounding parenchyma. However, a minority of benign polyps show a slight and delayed hypo-enhancement (unlike the more pronounced wash-out of malignant lesions) [2,13,14,32,100,113–115].

There is some evidence that high-frame-rate CEUS may better differentiate between cholesterol and adenomatous polyps [116].

Recently, some studies have shown that MVFI can differentiate between benign, adenomatous and malignant polyps [20,117].

Tridimensional ultrasound (3DUS) examination has also been proposed to better define gallbladder polyps [118].

Artificial intelligence, in particular radiomic analysis based on B-mode and SMI, has been applied in the sonographic differentiation between neoplastic and cholesterol polyps [119].

In recent years, some authors have developed scoring systems to predict the histologic type of gallbladder polyps based on pre-operative sonographic findings and/or patients characteristics [120,121]. Polyps > 10 mm in size have been considered as preinvasive adenomas or papillary neoplasms, while polyps from 6 to 10 mm in size rarely progress to malignancy [3,32]. For this reason, cholecystectomy has been proposed for all the polypoid lesions larger than 10 mm, while for polyps smaller than 10 mm, a simple follow-up has been suggested. Recently, concomitant risk factors for gallbladder malignancy have been taken into account: elderly age (more than 60 years), Asian ethnicity (especially Indian), a history of primary sclerosing cholangitis and concomitant focal gallbladder wall thickening (>4 mm) [122]. According to recent guidelines, the sonographic follow-up in patients without additional risk factors for malignancy should be related to the polyp size [123].

## 11. Gallbladder Carcinoma

GBC is the most common malignant neoplasm of the biliary tract, [124] especially in Asian countries [125]. It is found incidentally at the histological examination in 1% of all cholecystectomy specimens [126]. GBC usually develops from underlying chronic cholecystitis, especially porcelain gallbladder. Gallstones represent a risk factor according to some authors, especially if they are long-standing and large-sized (>3 cm) [127]. Symptoms are often late and unspecific (e.g., right upper quadrant pain, jaundice and weight loss) [124]. In fact, GBC is an incidental finding in almost 50% of cases, usually diagnosed via post-cholecystectomy histology [128]. The most frequent site for GBC is the fundus of the gallbladder (about 60%), followed by the body (30%) and the neck (10%) [129]. Very rarely, GBC arises from cystic duct [125].

US usually represents the first initial assessment in symptomatic patients, as it has a high accuracy (more than 80%), especially in the diagnosis of locally advanced GBC. However, US examination has a poor sensitivity in detecting early GBC [3,130]. GBC can show different patterns: a mass obscuring or replacing the gallbladder, a polypoid mass, a focal or diffuse thickening or an irregularity of the gallbladder wall. Intraluminal gallstones are usually found [1].

A mass obscuring or replacing the gallbladder is the most common presentation of GBC. At US examination, GBC appears to be an ill-defined mass in the gallbladder fossa, being mainly hypo- or isoechoic to the liver. The echotexture is usually mixed and dishomo-geneous, sometimes with anechoic regions (tissue necrosis or, less frequently, residual bile) or intralesional gallstones (intralesional hyperechoic foci). Generally, the GBC shape is

not well defined, and the margins can be irregular due to the invasion of the adjacent hepatic parenchyma. Notably, the gallbladder can be partially or completely masked or replaced by the malignant mass (Figure 6) [131]. Small amounts of pericholecystic fluid can be detectable via US, suggesting a poor prognosis [1,132].

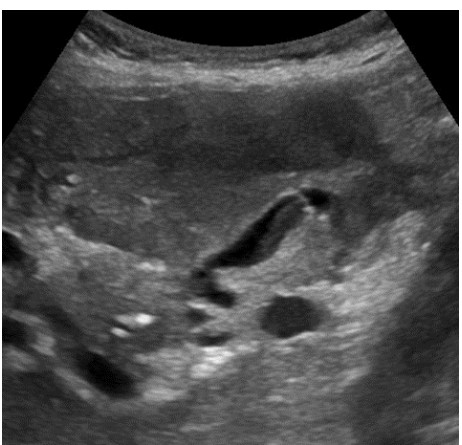

**Figure 6.** Gallbladder carcinoma. The gallbladder fossa is replaced by an ill-defined and heterogenous mass, infiltrating the liver parenchyma. The gallbladder is partially masked by the malignant mass, and the residual lumen, containing a small amount of biliary sludge, can be recognized.

The histological examination of a polypoid mass of the gallbladder wall can reveal an early-stage GBC, well differentiated and confined to the mucosa or muscularis propria [132]. At US examination, the polypoid mass can be hyper-, hypo- or isoechoic to the liver, usually with a homogeneous tissue texture and without acoustic shadow. A polypoid GBC generally has a large implant and smooth borders [123].

The rare GBC variant presenting as a focal or diffuse thickening or irregularity of the gallbladder wall is generally difficult to diagnose [1]. GBC confined to gallbladder mucosa may present as a flat or slightly raised lesion with luminal surface irregularity, sometimes without an appreciable wall thickening [1]. At more advanced stages, GBC can show marked mural thickening, often with irregular and mixed echogenicity [1]. Some authors described a thicker (>10 mm) and less echoic gallbladder wall in GBC compared to chronic cholecystitis [1,131].

*Sonographic Differentiation and Characterization of GBC*

Sonographic characterization and even differentiation between benign and malignant gallbladder lesions can be challenging.

At B-mode evaluation, a larger lesion and disrupted gallbladder wall suggest malignancy [13]. However, only coexistent signs of pericholecystic invasion are reliable indicators of GBC, such as lymphoadenopathies, the vascular infiltration of the hepatic pedicle, peritoneal involvement, liver metastases and pericholecystic fluid [131].

HRUS may be useful in differentiating GBC from benign lesions, especially GA and XGC. In particular, some studies indicate that HRUS can differentiate GA from early-stage GBC, with a diagnostic performance comparable to MRI and magnetic resonance cholangiopancreatography [79,133,134].

Sometimes, CD analysis can help in the diagnosis of GBC. In case of detectable CD signal, a power Doppler blood flow higher than 20–30 cm/s suggests a malignant lesion [130,135].

CEUS can be useful in differentiating GBC from benign lesions, mainly tumefactive sludge and chronic cholecystitis [15,32]. Arterial hyper-enhancement is not per se diagnostic because it is also common in benign gallbladder lesions [32]. However, during the arterial phase, the detection of linear or irregular and branched vessels can suggest malignancy [13,136,137]. An early wash out (within 35 s) is strongly suggestive of malignancy [13,32,137]. In fact,

about 91% of GBC and only 17% of benign lesions display wash-out within 35 s [134]. At CEUS examination, the disruption of the gallbladder wall and infiltration of the adjacent liver tissue are considered accurate features of malignancy [13,32]. The CEUS sensitivity for GBC is significantly lower for lesions ≤ 1 cm, [16,138] while for larger lesions, its performance seems to be comparable to those of CT and MRI [139]. Recently, promising data have been proposed for the diagnostic accuracy of some parameters derived from the wash in/wash out curve in the automated CEUS quantitative analysis (e.g., rise time, mean transit time, time to peak, fall time) [13,140]. Of note, according to some authors, CEUS performed with high frequency linear transducers could be a useful tool in GBC diagnosis, especially in the case of focal fundal gallbladder wall thickening [141].

At MVFI evaluation, malignant lesions can display tortuous micro-vessels or abrupt vascular caliber changes [20].

3DUS may be useful in the characterization of a gallbladder mass suspect for malignancy to better evaluate its location and extension [118].

Elastography has also been proposed for the evaluation of GBC [118].

## 12. Gallbladder Metastases from Different Organ Neoplasia

Metastatic diffusion to the gallbladder wall is rare, accounting for less than 5% of all gallbladder malignancies. In more than 50%, the primitive tumor is melanoma, followed by breast cancer, hepatocellular carcinoma, renal cell carcinoma and gastrointestinal tract cancers [142,143]. Gallbladder metastases are usually asymptomatic, but, in some cases, they can manifest as AC, especially in cases of larger polypoid lesions or the involvement of the neck or the infundibulum [144,145].

At B-mode evaluation, gallbladder metastases usually appear as single or multiple intraluminal nodules [32]. Occasionally, they present as focal or diffuse wall thickening [144,146]. Metastatic lesions from melanoma typically display a low-to-moderate echogenicity, probably due to the low reflectivity of melanin [146]. Unlike primary gallbladder adenocarcinoma, secondary neoplastic lesions of the gallbladder are rarely associated with gallstones [147]. CD evaluation may show intralesional signals, although the absence of vascular signals does not exclude malignancy [147–149].

Secondary gallbladder localization of different neoplastic lesions may show a vascular signal at CEUS examination, sometimes with arterial hyper-enhancement and wash out in the late phase [32,146,147].

## 13. Rare Gallbladder Neoplasia

Occasionally, gallbladder can be affected by lymphomas and neuroendocrine tumors that generally show the same sonographic features of GBC (see the specific section) [150–152].

Gallbladder cystoadenoma and mucinous cystic neoplasm appear at US evaluation as a cystic lesion, more often multiloculated. In most cases, the internal structure has a serous anechoic content, sometimes with the presence of echogenic material, suggestive of blood, mucin or protein aggregates [153,154]. Walls and septa can be thickened, with calcifications, casting posterior acoustic shadows. Papillary projections and nodular solid components have also been described [155,156].

## 14. Heterotopic Tissue in the Gallbladder

Heterotopia (normal tissue in an abnormal location) is rarely found in the gallbladder [157]. Most frequently, it is represented by gastric mucosa, followed by pancreatic and, very rarely, thyroid, liver and adrenal gland tissue [157]. The most frequently involved portions are the cystic duct and the gallbladder neck [158].

At US examination, heterotopic tissue in the gallbladder appears mainly as a polypoid mass, either sessile or pedunculated, more often hyperechoic or, sometimes, isoechoic, as well as of variable size (from 5 to 30 mm). Less frequently, it presents as focal gallbladder wall thickening, usually hyperechoic [158]. Occasionally, the mucus secretion of heterotopic gastric or pancreatic tissue can lead to the development of a parietal cystic lesion [159,160].

## 15. Gallbladder Trauma

Gallbladder injury occurs in about 2% of blunt abdominal trauma cases and is usually associated with the involvement of the liver or different abdominal organs [161]. Isolated gallbladder injury is even rarer, due to its size and its localization, as it protected by the liver and the ribs. Injury of the gallbladder includes contusion (intramural hematoma), perforation and avulsion [162].

At US examination, the finding of discontinuous or irregular gallbladder wall is suggestive of perforation, especially in presence of a collapsed gallbladder [161,163]. The findings of echoic material within the gallbladder lumen should raise suspicion of intraluminal bleeding [164,165]. Pericholecystic or perihepatic fluid and gallbladder wall thickening represent common but less specific findings [161,164].

## 16. Gallbladder Edema

Besides typical calculous and acalculous AC, different systemic and local diseases can determine gallbladder wall edema. Among these, the most frequent are hepatic cirrhosis (Figure 7), acute hepatitis, acute pancreatitis, congestive heart failure, hypoalbuminemia, sepsis, acute renal failure, peritonitis and pyelonephritis [2,166].

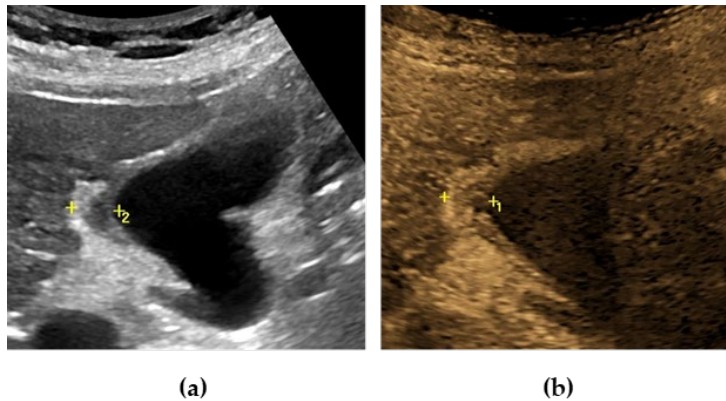

(a)                                          (b)

**Figure 7.** Gallbladder edema due to hypertensive cholecystopathy in a patient with hepatic cirrhosis. At B-mode evaluation, gallbladder walls are thickened with a double-wall appearance (calipers) (**a**). The same case evaluated via MVFI, which depicts the vascular pattern of the thickened gallbladder wall (calipers) with pronounced higher vascular signals in the outer layer and an edematous hypovascular inner layer (**b**).

Notably, in bone marrow transplant patients, gallbladder edema is important for the suspicion of sinusoidal obstruction syndrome (SOS), also known as veno-occlusive disease (VOD) or hepato-biliary graft-versus-host-disease (GVHD) (Figure 8) [167,168].

Gallbladder edema has been also described in course of salmonella enteric infection, molecular targeted therapies, hyperthyroidism, pre-eclampsia, Dengue fever, gold salt hypersensitivity, acute peri-myocarditis, scarlet fever and COVID-19 infection [169–177].

In case of acute hepatitis, especially viral etiology, gallbladder wall edema is often associated with biliary sludge, hepatomegaly and the diffusely hypoechoic texture of the liver, sometimes with prominent portal triads ("starry sky" appearance) [178].

The sonographic hallmark of gallbladder edema is the presence of wall thickening (>3 mm) with a multilayered and meshwork pattern [166]. A double-wall appearance can be seen, characterized by the presence of hyperechoic outer and inner borders with a relatively narrower echogenic layer in between [178]. Gallbladder edema can be misdiagnosed with the pseudo-thickening observed in the post-prandial state due to the gallbladder physiologic contraction [166].

CEUS could be useful to differentiate gallbladder wall edema from gallbladder wall thickening due to AC. In case of simple edema, the contrast enhancement is observed in

the inner and outer layers of the gallbladder wall but not in the hypoechogenic edematous area in between [14].

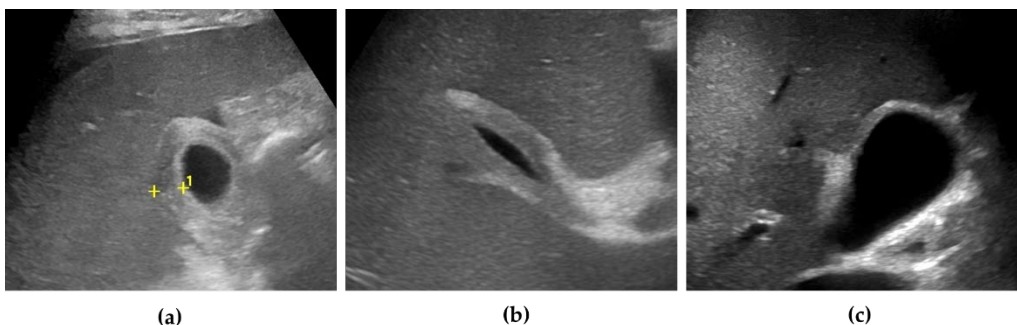

(a)                                      (b)                                      (c)

**Figure 8.** Gallbladder edema evolution in a young patient with VOD after bone marrow transplant, before and after treatment. At diagnosis, gallbladder walls appear markedly thickened. A small amount of perihepatic and pericholecystic fluid can be seen (**a**). After a few days of treatment with defibrotide, a progressive decrease in the gallbladder wall was observed, with disappearance of pericholecystic fluid (**b**,**c**).

## 17. Gallbladder Volvulus

Gallbladder volvulus (GV), also known as gallbladder torsion, is a rare condition, accounting less than 500 cases in the literature. GV consists of the rotation of the gallbladder on its long axis, resulting in impaired vascular supply with subsequent gallbladder wall ischemia [179]. GV typically occurs in the elderly females or, less frequently, in pediatric patients and young adults. Patients with complete torsion (>180°) usually present a clinical picture resembling AC. Differently, in cases of incomplete gallbladder torsion (<180°), patients may experience recurrent episodes of slowly progressive biliary pain [179,180].

At US examination, the most common finding of GV is the presence of a "floating gallbladder" with a thickened wall [179]. US examination may show a hypoechoic edematous layer between the muscular wall and the mucosa due to venous and lymphatic stasis. Some authors have reported the sonographic finding of a stretched cystic duct and gallbladder neck, appearing as a conical-shaped structure composed of multiple linear echoes converging towards the tip ("cystic duct knot sign") [181–183]. The sonographic finding of portal venous gas associated with gallbladder ischemia in GV has been described [184].

CD evaluation can detect blood flow interruption in the cystic pedicle, typically associated with complete gallbladder torsion and the absence of vascular signals in the gallbladder wall [179,180].

A sonographic picture suggestive of GV strongly indicates an emergency cholecystectomy [179].

## 18. Hemobilia

Haemobilia is defined as the presence of macroscopic endoluminal blood in the biliary tree or in the gallbladder [185]. Haemobilia is most frequently secondary to complications of invasive procedures on the hepatopancreatobiliary system [185]. Additional causes of haemobilia include traumatic injury, biliary tumors, inflammatory disease, venous or arterial-biliary fistulae, rupture of aneurysm of the hepatic artery, hemorrhagic cholecystitis and coagulation disorders [3]. Concomitant right upper quadrant pain with jaundice and overt gastrointestinal bleeding (Quinckle's triad) has been classically described [44].

At US examination, blood initially appears as an echoic material in the gallbladder lumen, which tends to layer in the most declivous portion [3,39,44]. Later in time, when the hematoma has developed, it looks like an echoic, usually dishomogeneous, nonmobile intraluminal mass, without acoustic posterior shadowing [3,44]. Rarely, intraluminal blood may develop a cystic appearance [39].

CEUS can be useful in the detection of active intraluminal gallbladder bleeding [14].

Sometimes, US evaluation can detect the underlying cause of hemobilia. In fact, some authors reported the rupture of artery pseudoaneurysm, appearing as a hypoechoic intraluminal mass of the gallbladder, with the typical "yin and yang" sign at CD examination. Power Doppler analysis shows a pulsatile wave pattern [186].

## 19. Gallbladder Ascariasis

Ascariasis is a human infestation caused by *Ascaris lumbricoides*, diffused in tropical and subtropical areas. *Ascaris lumbricoides* adult worms can migrate from the duodenum to the biliary tract and into the gallbladder [187].

At US examination, active live worms in the gallbladder appear as long, curved or coiled up echoic structures, with active movements and without acoustic posterior shadow. They can show a longitudinal central anechoic tube, representing the digestive tract of the worm, surrounded by a thick echoic stripe ("inner tube sign" or "triple line sign"). Alternatively, they can simply appear as a thin tubular echoic stripe ("stripe sign") [188]. In the transversal scan, worms display a round-shaped structure, with a target-like appearance [189]. Hook-shaped and coiled worms can lead to the septated appearance of the gallbladder or, sometimes, a "bull's eye" sonographic picture [187,190]. In case of living worms, US is the first-choice technique for the diagnosis of gallbladder ascariasis (84% sensitivity) [191,192]. Notably, the sonographic finding of non-movable worms is suggestive of their death or paralysis.

Some different worms (e.g., *Clonorchis sinensis*, *Opisthorchis viverrini*, *Opisthorchis felineus* and *Fasciola hepatica*) can rarely lead to a similar sonographic picture [193].

## 20. Congenital Variants of the Gallbladder

Congenital variants of the gallbladder represent a rare entity, and, sometimes, they are an incidental sonographic finding. Their early recognition avoids misdiagnosis and unnecessary diagnostic workup. Notably, 3DUS can be useful to highlight the anatomical features of gallbladder congenital variants [118].

The gallbladder is a pear-shaped organ, but different morphologies are described. The most frequent variants are the Phrygian cap gallbladder (the fundus is folded over the body of the organ) and the sigmoid gallbladder, appearing as a double cavity because of a septum or a folded morphology [194].

Further congenital gallbladder variants are as follows:

- Agenesis of the gallbladder. It is defined as the absence of the gallbladder in patients without a history of cholecystectomy. This rare condition (10 to 65 per 100,000) is frequent in patients with biliary atresia (1 out of 6 patients). In almost half of patients, agenesis of the gallbladder is associated with the development of common duct stones. It is important to rule out gallbladder ectopia [195].
- Hypoplasia of the gallbladder. In adults, the minimal gallbladder length is generally 7 cm, and the minimal width is 2 cm [195]. Researchers have reported associations with cystic fibrosis, cholangitis and biliary atresia. At US examination, the hypoplastic gallbladder seems to be contracted, collapsed or simply small in size. The microgallbladder, typically associated with cystic fibrosis, is defined as a gallbladder < 2–3 cm long and < 0.5–1.5 wide [196].
- Septated and multiseptated gallbladder. Very rarely, US shows a multichambered or multiloculated gallbladder lumen with multiple thin intraluminal septa, sometimes with a "honeycomb" appearance [195].
- Duplicated gallbladder. The duplication of the gallbladder consists of the presence of two completely separated gallbladder cavities that can present a common cystic duct (bilobed gallbladder) or two different cystic ducts. In the latter case, the two separated cystic ducts may have a common insertion in the main bile duct (Y-shaped gallbladder) or, alternatively, two distinct insertion points (V-shaped gallbladder) [195]. The presence of sludge, cholelithiasis or disease—for example, cholecystitis—in only a

single gallbladder cavity helps to detect the presence of two different gallbladder lumens, suggesting the diagnosis of a duplicated gallbladder [195].

- Intrahepatic gallbladder. US shows the gallbladder partially embedded or completely incorporated into the hepatic parenchyma. This anatomic variant is usually associated with biliary stasis because of ineffective gallbladder emptying, as well as with an increased risk of torsion [195].

- Left-sided gallbladder. In this ectopic variant, the gallbladder is located on the left side of the *ligamentum teres* between the segments III and IV or on segment III. It can be associated with *situs viscerum inversus*, portal vein or biliary system anomalies and segment IV atrophy. At US examination, the left-sided gallbladder generally appears as a cystic mass near the left lobe of the liver, in front of the pancreas, with a narrow neck connecting to the bile duct [195,197].

- Rarer variants of ectopic gallbladder. Rarer variants of ectopic gallbladder have been described in the literature, namely retrohepatic, suprahepatic, supradiaphragmatic, retroperitoneal, intrathoracic, within the falciform ligament or within the abdominal wall musculature. The sonographic visualization of gallbladder may be particularly challenging for these rare variants [195].

## 21. Gallbladder Dysmotility

Gallbladder dysmotility is a functional gallbladder disorder, characterized by biliary pain in the absence of gallstones, sludge or structural disease [198]. Gallbladder dysmotility can be associated with different conditions, such as diabetes, obesity, myotonic dystrophy, cirrhosis, irritable bowel disease, slow transit constipation, medications and celiac disease [199]. Additionally, in patients with celiac disease, an enlarged gallbladder, often containing sludge, can be found at US examination [200].

The finding of a low ejection fraction, evaluated via cholecystokinine-stimulated cholescintigraphy (HIDA scan), supports the diagnosis of gallbladder dysmotility [201]. Similarly, US is used for the evaluation of gallbladder motility, both in research studies and some clinical settings [202,203]. The gallbladder functional sonographic study is based on the calculation of its volume at the baseline and then at regular intervals after the ingestion of a standard fat meal in order to evaluate gallbladder emptying and subsequent filling. A software has been developed to calculate the gallbladder volume after the acquisition of two-dimensional sonographic imaging [204,205]. Recently, 3DUS and 4DUS (i.e., dynamic 3DUS) have improved the evaluation of gallbladder motility, with a diagnostic performance similar to cholescintigraphy [206].

## 22. Conclusions

US is a highly effective imaging tool for the diagnosis of gallbladder disease. Since its introduction into medical practice, US has acquired much importance not only in the elective setting but also as a prompt confirmation of clinical suspicion in patients with right upper quadrant pain or possible gallbladder disease. In particular, gallbladder POCUS can be easily performed by the clinician in a medical office or an emergency room.

Recently, new sonographic tools have been developed, such as HRUS, MVFI/SMI, CEUS, 3D-US, elastography and artificial intelligence-powered US. These new US-based techniques will need further evaluation to elucidate their individual diagnostic potential. Importantly, the combination of multiple sonographic tools (so-called multiparametric US) can improve the diagnostic yield of US examination.

**Author Contributions:** Conceptualization, L.M. and M.M.; methodology, L.M. and M.M.; software, L.M. and M.M.; validation, L.M., A.V. and M.M.; formal analysis, L.M. and M.M.; investigation, L.M. and M.M.; resources, L.M., A.V., R.M.Z. and M.M.; data curation, L.M. and M.M.; writing—original draft preparation, L.M. and M.M.; writing—review and editing, L.M., A.V., R.M.Z. and M.M.; visualization, L.M. and M.M.; supervision, R.M.Z.; project administration, R.M.Z.; funding acquisition, R.M.Z. All authors have read and agreed to the published version of the manuscript.

**Funding:** This research received no external funding.

**Conflicts of Interest:** The authors declare no conflicts of interest.

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
