# Peer review of "New Developments in the Ultrasonography Diagnosis of Gallbladder Diseases"

_gastroent, doi:10.3390/gastroent15010004_

Round 1
Reviewer 1 Report
Comments and Suggestions for Authors
Good paper, only a comment, the point is after the reference, not before. Please change it in all the paper. And the title of review is a little too long.
This paper is an extensive overview about the value of US in gallbladder pathology. The paper is not original, is a “classical”one, but is wellcome for readers! It is nothing special new in the paper, but is extensively presented the gallbladdre pathology and the rple of US for diagnosis!
Author Response
Author’s point by point reply to Reviewer 1 comments and suggestions
- Good paper, only a comment, the point is after the reference, not before. Please change it in all the paper. And the title of review is a little too long.
Author’s reply:
- We have switched the position of the references according to the Reviewer’s suggestion.
- We propose a shorter title for the paper: “New developments in the ultrasonography diagnosis of gallbladder diseases.”
- This paper is an extensive overview about the value of US in gallbladder pathology. The paper is not original, is a “classical” one, but is welcome for readers! It is nothing special new in the paper, but is extensively presented the gallbladder pathology and the role of US for diagnosis!
Reviewer 2 Report
Comments and Suggestions for Authors
The paper is well written and exaustive. It is a detalied review of the current indications of the applications of US in gallbladder disease.
I commend the authors for putting together such a complete review.
I am not sure this is fitting with the journal requirements. this looks more like a book chapter and or a monography on the subject with 20 pages and 250 references.
I will defer to the Editors to confirm the suitability of this work to the Journal.
Author Response
- The paper is well written and exhaustive. It is a detailed review of the current indications of the applications of US in gallbladder disease.
- I commend the authors for putting together such a complete review.
- I am not sure this is fitting with the journal requirements. This looks more like a book chapter and or a monograph on the subject with 20 pages and 250 references.
Author’s reply:
- This article is aimed at a medical audience interested in the US diagnostic approach to gallbladder diseases, both for initial diagnosis and for clinical follow-up, with particular attention to new technical advances. In recent years, the medical community has paid much attention to optimizing the management of gallbladder diseases, due to their high prevalence and clinical importance. Therefore, knowledge of gallbladder diseases and the related US findings is very important for both the clinician and the radiologist.
- According to the Reviewer’s suggestion, we have reduced the number of references, eliminating those that are not strictly necessary.
- I will defer to the Editors to confirm the suitability of this work to the Journal.
Round 2
Reviewer 2 Report
Comments and Suggestions for Authors
Thank you